# Dough Rheological Properties, Microstructure and Bread Quality of Wheat-Germinated Bean Composite Flour

**DOI:** 10.3390/foods10071542

**Published:** 2021-07-03

**Authors:** Denisa Atudorei, Olivia Atudorei, Georgiana Gabriela Codină

**Affiliations:** Faculty of Food Engineering, Stefan cel Mare University of Suceava, 720229 Suceava, Romania; denisa.atudorei@outlook.com (D.A.); olivia.atudorei@outlook.com (O.A.)

**Keywords:** germinated bean flour, refined wheat flour, dough rheology, bread quality

## Abstract

Germinated bean flour (GBF) was obtained and incorporated in different levels (5%, 10%, 15%, 20% and 25%) into dough and bread made from refined wheat flour. The incorporation of GBF into wheat flour led to a decrease of the water absorption value, dough consistency, baking strength, extensibility and improved tolerance for mixing, total gas production and α-amylase activity. Tan *δ* increased in a frequency-dependent manner for the samples with a GBF addition, whereas the *G’* and *G”* decreased with the increased value of the temperature. According to the microscopic structures of the dough samples, a decrease of the starch area may be clearly seen for the samples with high levels of GBF addition in wheat flour. The bread evaluation showed that the specific volume, porosity and elasticity increased, whereas the firmness, gumminess and chewiness decreased up to a level of 15% GBF addition in wheat flour. The color parameters *L**, *a** and *b** of the bread samples indicated a darkening effect of GBF on the crumb and crust. From the sensory point of view, the bread up to a 15% GBF addition was well-appreciated by the panelists. According to the data obtained, GBF could be recommended for use as an improver, especially up to a level of 15% addition in the bread-making industry.

## 1. Introduction

Bread is one of the most consumed foods in the world. Nowadays, we are trying to diversify the bread products in order to comply with consumer demands [1]. Additionally, it is of a particular interest in improving bakery products from a nutritional and technological point of view without any chemical additive additions that could endanger the health of consumers [2]. A possibility for improving the bread quality is to use germinated legume grains in bread making [3,4,5,6,7]. This is due to the advantages of the germination process [8,9], which increases the bioavailability of nutrients due to the fact that some compounds are broken down in small components that become easier to digest and to be absorbed by the human body [10,11,12,13]. Additionally, germination increases the amount of desirable compounds, such as phenolic ones [14], and some minerals, such as calcium, magnesium, zinc and iron [15,16,17,18], decrease the amount of antinutritive factors such as phytic acid that combine with minerals and result in phytates [19,20,21,22]. Through germination, the enzymatic activity of grains is increased, which has an effect in facilitating the digestion of compounds such as starch and proteins and, therefore, enables germinated legumes to be used successfully in foods where enzyme activity is required. The natural deficiency in wheat of some enzymes requires their addition to wheat flour to improve the rheological properties of the dough and the quality of bakery products. Among the enzymes with an essential role in baking are amylases, which are most often found in insufficient amounts in flour from ungerminated wheat. In baking, these enzymes play a double role. On the one hand, they continuously supply fermentable carbohydrates for yeasts and, in this way, ensure the continuous production of carbon dioxide, and on the other hand, they contribute to the improvement of the properties of the dough (decrease in viscosity and dough consistency) and, therefore, to the high quality of the bread [23]. The importance of adjusting the level of amylase activity of wheat flours is essential for the dough fermentation process. Wheat flour contains small amounts of fermentable carbohydrates. Amylases act on starch, which is found in a large amount in wheat flour, with maltose formation used by yeast for the production of carbon dioxide, ethanol and other fermentation products, which leads to a rapid and uniform dough fermentation [24].

However, the incorporation of germinated legumes grains must be carefully chosen by taking into account the sensory preferences of consumers, the nutritional profile of legumes grains and their impact on bread from a technological point of view. In this sense, beans, a legume that is highlighted by a nutritional profile appreciated by nutritionists, are of particular interest for use in bread making.

Borlotti beans belong to the family *Fabaceae*, genus *Phaseolus*, species *vulgaris*. Beans are of particular nutritional interest, because they are an important source of protein and carbohydrates [25], phenolic compounds [26] and have positive implications for the prevention and treatment of diseases, such as: colon cancer [27], diabetes [28], cardiovascular diseases [29] and metabolic diseases [30]. Beans are one of the most important sources of vegetable proteins, with a content of 15–25% protein and considered the meat of poor people, because they are the cheapest source of protein. Additionally, the amount of protein in beans, along with color, is an indicator of their quality [31]. Studies show that beans are among the only foods of plant origin that provide significant amounts of the indispensable amino acid lysine [32], which is deficient in wheat flour [33]. In the category of amino acids, present in significant amounts in beans are also glutamic acid, aspartic acid, serine, arginine, alanine, threonine, leucine and phenylalanine [34]. Beans also present a high content of bioactive substances, such as tannins, flavonoids (phenolic compounds) and other antioxidants [25,32,35]. They contain a significant amount of complex carbohydrates, the values of which range between 52% and76%. Carbohydrates are composed of starch, which represents more than 50% of the seed’s weight; 14–19% fiber, of which fifty percent are in a soluble form, and lower amounts of mono-, di- and oligosaccharides. The lipid contents of beans range between 1.5% and 6.2%, consisting of acylglyceride mixtures in which the main fatty acids are mono- and polyunsaturated [36]. The nutritional value of beans is limited by the presence of some antinutrients such as saponins, lectins, phytate, trypsin inhibitors and tannins, which can affect the protein digestibility or mineral bioavailability and, therefore, its usage by the human body [37]. Through germination, the nutritional profile of beans is improved. This process increases the bioavailability of the nutrients, such as amino acids like lysine, which is deficient in wheat flour [33]; minerals present in the beans, such as magnesium, iron, calcium and vitamins, especially the B complex, in which beans are rich [38], decreases the amount of antinutrient compounds, such as phytic acid [39], trypsin inhibitors [37], and activates hydrolytic enzymes in the grain [40].

However, there are few studies made on the possibility of using germinated beans in a bread-making recipe. This study presents in detail the effect of the addition of germinated bean flour on the rheological properties of dough, its microstructure and on the quality of the bread. All these aspects need to be highlighted in order to determine exactly the optimal amount of germinated bean flour in a refined wheat bread production recipe.

## 2. Materials and Methods

### 2.1. Materials

Commercial refined wheat flour was used. Wheat flour type 650 was bought from the S.C. Dizing S.R.L. company (Brusturi, Neamț, Romania). Germinated bean flour was obtained from beans (*Phaseolus vulgaris*) cultivated in Suceava County, Romania. The beans were germinated, lyophilized and milled before they were used in the wheat flour. The germination was made in dark conditions at a temperature of 25 °C and a constant humidity of 80%, according to the methods reported in our previous studies [41,42]. After 4 days of germination, the bean seeds were freeze-dried in a lyophilizer (Biobase, BK-FD12, (Jinan, China), taking into account the following parameters: temperature −50 °C, 24 h and a pressure of 10 Pa. After lyophilization, the bean seeds were milled in a laboratory mill 3100 (Perten Instruments, Hägersten, Sweden) in order to be added into wheat flour. The lyophilization process was used, to the detriment of the drying methods of beans that involve high temperatures, in order to better preserve some nutritional and bioactive compounds in the grains but, also, the enzymes that were activated during germination to protect the unstable compounds at high temperatures [43,44].

The raw materials and bread samples were analyzed according to ICC standard methods: moisture content (ICC 110/1), ash content (ICC 104/1), protein content (ICC 105/2) and fat content (ICC 136). To be certain that the bean flour after germination and lyophilization can be incorporated into the purchased wheat flour, this was analyzed through a microbiological point of view as follows: yeast and molds according to SR ISO 7954:2001, *Bacillus cereus* according to SR EN ISO 7932-2003:2005 and mycotoxins by using an ELISA kit (Prognosis Biotech, Larissa, Greece). The wheat flour was analyzed for the gluten deformation index and wet gluten content according to Romanian standard SR 90/2007. The falling number value of the wheat flour was analyzed according to the ICC 107/1 method.

### 2.2. Dough Rheological Properties

#### 2.2.1. Dough Rheological Properties during Mixing and Extension

An Alveo Consistograph (Chopin Technologies, Cedex, France) was used to determine the dough rheological properties during mixing (Consistograph test) and extension (Alveograph test). The Consistograph test was made according to ICC 171 and AACC 54–50 approval. The Alveograph test was done according to ICC 121, AACC 54–30A and ISO 5530/4 approval at a 14% moisture basis at constant hydration. The Consistograph test at a 14% moisture basis determined the water absorption capacity (WA), maximum pressure (PrMax), tolerance to kneading (Tol), consistency of the dough after 250 s (D250) and consistency of the dough after 450 s (D450). The Alveograph test determined the maximum pressure (P), dough extensibility (L), index of swelling (G), baking strength (W) and configuration ratio of the Alveograph curve (P/L).

#### 2.2.2. Dough Rheological Properties during Fermentation and Falling Number Values

The dough rheological properties during fermentation used a Rheofermentometer device (Chopin Rheo, type F3, Villeneuve-La-GarenneCedex, France) according to AACC89–01.01 approval. The falling number values were made using a falling number device (FN 1305, Perten Instruments AB Stockholm, Sweden) according to ICC 107/1 approval. The Rheofermentometer test determined the total CO_2_ volume production (VT, mL), maximum height of gaseous production (H’m, mm), volume of the gas retained in the dough at the end of the test (VR, mL) and retention coefficient (CR, %). For this purpose, dough samples were obtained by kneading 250-g mixed flours, 5-g salt and 7-g compressed yeast of the *Saccharomyces cerevisiae* type, according to the Consistograph water absorption value. The falling number test determined the falling number value (FN, s).

#### 2.2.3. Dough Fundamental Rheological Properties

The dough fundamental rheological properties were made using a HAAKE MARS 40 rheometer device (Termo-HAAKE, Karlsruhe, Germany) with a plate and plate system of 40 mm in diameter and a gap of 2 mm. The dough samples were mixing at the optimum Consistograph water absorption by using the Alveo Consistograph and then placed between rheometer plates and rested before analysis 5 min for relaxation. Frequency sweep tests from 1 to 20 Hz were performed at 25 °C in a range of linear viscoelasticity previously established. The storage modulus (*G’*), loss modulus (*G”*) and loss tangent (tan *δ*) were carried out at a constant stress of 15 Pa for the frequency sweep tests and during heating from 25 to 100 °C at a heating rate of 4 °C per min at a frequency of 1 Hz and a fixed strain of 0.001.

### 2.3. Dough Microstructure

The epifluorescence light microscopy (EFLM) images of dough with different levels of germinated bean flour addition were observed at room temperature with a Motic AE 31 (Motic, Optic Industrial Group, Xiamen, PR China) equipped with catadioptric objectives LWD PH 203 (N.A. 0.4). The dough samples were prepared, and the images were obtained and analyzed according to methods reported in our previous studies [45,46].

### 2.4. Bread Making

The bread samples were obtained through the following steps: dosing the ingredients (wheat flour and GBF, which substituted wheat flour in different levels of 5%, 10%, 15%, 20% and 25% of salt, yeast and water); mixing them for 15 min in in a heavy duty mixer (Kitchen Aid, Whirlpool Corporation, Benton Harbor, MI, USA), divided in three pieces of 400 g each; leavening for fermentation for 60 min at 30 °C in a fermentation chamber (PL2008, Piron, Italy) and baking the dough samples for 30 min to 220 °C in an electrical bakery convection oven with steam production, ventilation and humidification (PF8004 D, Piron, Italy). The ingredients used in the sample production were: white wheat flour type 650; germinated bean flour (in variable proportions of 5%, 10%, 15%, 20% and 25%); compressed yeast of the *Saccharomyces cerevisiae* type (3%); sodium chloride (1.5%) and distilled water, according to the water absorption capacity value of wheat-germinated bean flours.

### 2.5. Bread Quality Evaluation

#### 2.5.1. Bread Physical Characteristics

The specific volume (seed displacement method), porosity and elasticity were determined in accordance with the SR 90: 2007 standard method.

#### 2.5.2. Color Parameters

Crumb and crust colors of the bread samples were made using the Konica Minolta CR-400 colorimeter (Tokyo, Japan). The parameters *L** (darkness/brightness), *a** (shade of red/green) and *b** (shade of blue/yellow) were analyzed. The determination was based on the CIE Lab* color system. The field in which the absorption of electromagnetic radiation was achieved was UV-VIS.

#### 2.5.3. Texture Profile Analysis

The texture characteristics of the bread sample were determined using the texturometer device TVT-6700 (Perten Instruments, Hägersten, Sweden). This texturometer was equipped with a 10-kg load cell. The firmness, gumminess, cohesiveness, resilience and chewiness were determined. For this purpose, the bread samples were cut into 50-mm-high slices and subjected to two compression cycles up to 20% of their initial height. For this, a cylindrical probe of 45 mm was used at a speed of 1.0 mm/s, a trigger force of 5 g and a recovery period between compressions of 15 s.

#### 2.5.4. Crumb Microstructure

To highlight the microstructure of the crumb, the Motic SMZ-140 stereo microscope (Motic, Xiamen, China) with a 20x objective was used, to a resolution of 2048 × 1536 pixels.

#### 2.5.5. Sensory Analysis

For the sensory determinations, a 9-point hedonic scale was used, and the fallowing characteristics were evaluated: appearance, color, aroma, taste, smell, texture and global acceptability. The sensory characteristics of the bread samples were evaluated by a panel of 20 semi-trained judges.

### 2.6. Statistical Analysis

All data were expressed as the mean ± standard deviation. Statistical analysis was carried out with Statistical Package for Social Science statistical package (v.16, SPSS, Chicago, IL, USA). A one-way analysis of variance (ANOVA) with Tukey’s test was used to determine the significant differences at a 5% level.

## 3. Results

### 3.1. Flour Characteristics

The physical–chemical characteristics of the wheat flour used as the base material were the following: 14.6% moisture, 0.66% ash content, 12.3% protein, 1.12% fat, 30.4% wet gluten and 3-mm gluten deformation index. The falling number of the wheat flour was 356 s. According to the data obtained, the wheat flour was of a very strong quality for bread making and had a low α amylase activity.

The germinated bean flour presented the following physical–chemical characteristics: 10.1% moisture, 3.0% ash, 26% protein and 1.4% fat. According to the data obtained, the germinated bean flour presented a high protein content, the values being in agreement with those reported by Kassegn et al. [47] and Poblete et al. [48] for germinated beans. From a microbiological point of view, the germinated bean flour presented the fallowing values: yeast and molds 1 UFC/g, free of *Bacillus cereus*, aflatoxin less than 1.4 ppb, ochratoxin 32.62 ppb and zearalenone 96.38 ppb. According to the data obtained, the germinated bean flour was from the microbiological point of view in the limits range recommended by the European Union and, therefore, may be used as ingredients in food products [49,50].

### 3.2. Dough Rheological Properties during Mixing and Extension

#### 3.2.1. Dough Rheological Properties during Mixing and Extension

The Consistograph data are shown in Table 1. As it may be noticed, the water absorption value and dough consistency after 250 and 450 s decreased with the increased level of GBF addition in the wheat flour. Regarding the dough tolerance to mixing, its value increased for dough samples up to 15% GBF addition, followed by a slight decrease. However, all the dough samples with a GBF addition presented higher values for the tolerance to mixing compared to the control sample.

The Alveograph characteristics are shown in Table 2. As it may be seen, the dough tenacity and configuration ratio of the Alveograph curve increased, whereas the dough extensibility, index of swelling and baking strength decreased with the increased level of GBF addition in the wheat flour. These results indicate a weakening effect of GBF on the mixed dough, taking into account that the W value decreased approximately two times for the GBF_25 sample compared to the control one.

#### 3.2.2. Dough Rheological Properties during Fermentation and Falling Number Values

The Rheofermentometer characteristics and the falling number values are shown in Table 3. As it may be seen, the H’m, VT and VR Rheofermentometer values increased with the increase level of GBF addition up to 10% and then decreased, the lowest values being recorded for the GBF_25 dough sample. Contrarily, the retention coefficient value decreased up to a level of 10% GBF addition; after this level, the CR value increased. The falling number index value decreased with the increased level of GBF addition in the wheat flour. The wheat flour used as a base for the GBF additions presented a high FN value, being flour with a high α-amylase activity [23]. By GBF addition, the FN value of the mixed flours (wheat-germinated bean) decreased below the value of 280 s, the value of which the mixed flour presented a normal α-amylase activity [24].

#### 3.2.3. Dough Fundamental Rheological Properties

The frequency sweep tests are shown in Figure 1. As it may be seen, the storage modulus (*G’*), loss modulus (*G”*) and loss tangent (tan *δ*) strongly depend on the frequency. As it may be seen, the *G’* was higher than *G”* in all frequency ranges, indicating that the elastic properties of the dough samples were more prominent than the viscous ones. Tan *δ*, which is the ratio of viscous and elastic components of the dough, was lower than 1 for all the analyzed samples.

A temperature sweep of the dough mix rheological properties is shown in Figure 2, which presents the *G’*, *G”* and tan *δ* variations. As it may be seen, the *G’* and *G”* decreased up to a certain temperature due to protein denaturation, fallowed by an increase of their values due to the starch gelatinization process. After gelatinization, the starch hydrolysis reduced the dough consistency, which began to increase only after the α-amylase activity was inactivated.

### 3.3. Dough Microstructure

EFLM was used to investigate the dough samples with different levels of germinated bean flour additions (Figure 3). Rhodamine B and fluorescein (FITC)-labeled proteins and starch-staining proteins are in red and starch granules in green. In general, rhodamine B presents an affinity for protein-rich domains. Therefore, a higher level of the protein amount in the dough system will lead to a higher accumulation of rhodamine B in the protein phase due to the hydrophobic affinities [51]. The fluorescent green-colored areas indicate the presence of starch, whereas the red-colored areas indicate the presence of proteins.

### 3.4. Bread Quality Evaluation

#### 3.4.1. Bread Physical Characteristics

The bread physical characteristics are shown in Table 4. As it may be seen, the specific volumes of the bread samples, compared to the control one, increased for the samples with 5%, 10% and 15% germinated bean flour (GBF) additions in wheat flour. An addition of more than 15% GBF had the effect of reducing the value of the specific volume, which was even lower than the control sample. Thus, it can be concluded that, in order to obtain bread with an improved specific volume, a maximum of 15% germinated bean flour can be added to wheat flour. The addition of GBF in wheat flour also had an influence on the porosity of the bread samples. The porosity was improved due to the GBF addition, but when the level exceeded 20%, the porosity was negatively influenced. Regarding the elasticity of the bread samples, it can be concluded that a higher level of GBF addition of 20% or 25% decreased the value of the elasticity, compared with the control sample, without any GBF addition in the wheat flour.

All the bread physical characteristics (specific volume, porosity and elasticity) are of particular importance for the quality of the finished products, because they are related to consumer acceptability, so it is necessary to monitor the influence of the addition of GBF on their values. A GBF addition up to 15% had the effect of improving specific volume and porosity values. Improving the specific volume of bread is closely related to increasing the porosity, both of which are desirable and may be an indicator of the freshness of this food product [52]. The values of elasticity were not significantly (*p* < 0.05) different when low levels of GBF were added to the wheat flour and decreased when high levels of GBF were incorporated in the bread recipe.

#### 3.4.2. Color Parameters of Breads Samples

Table 5 shows how the color parameters of the crumb and crust of the bread samples *L**, *a** and *b** changed depending on the GBF addition level in the wheat flour. Evaluating the color parameters is important, because the color of the bread is a feature closely related to its quality [53]. The color of the bread samples was influenced by the ingredients used, their proportion in the manufacturing recipe and the baking parameters. The color values are largely influenced by the acceptability of consumers [54].

As it may be seen, the value of the parameter *L** (brightness) decreased due to the addition of germinated bean flour. Thus, it can be concluded that the bread samples with the addition of GBF had a lower brightness, both of the crust and of the crumbs. Regarding the parameter *a**, it was observed that the control sample had the lowest value, and then, it increased with the increase of the GBF addition level. Thus, it can be concluded that the GBF addition had the effect of increasing the red value of the samples. The color data obtained also showed that the value of yellow increased due to the GBF addition in the wheat flour. The *L**, *a** and *b** value trends were similar for both bread areas analyzed, namely the crust and crumbs.

#### 3.4.3. Texture Profile Analysis of Breads Samples

The textures of the bread samples are shown in Table 6. The bread texture is a very important characteristic, because it has a direct influence on consumer perceptions. The textural properties can be determined by measuring the resistance to deformation of a bread sample during the application of a force [55]. As it may be seen from Table 5, all the parameters that characterize the texture (firmness, gumminess, chewiness, cohesiveness and resilience) were influenced by the addition of germinated bean flour. It was observed that the value of the firmness parameter was much higher than that of the control sample after the addition of 20% GBF in wheat flour. Additionally, the gumminess and chewiness values, starting with an addition of 20% GBF in wheat flour, were significantly higher (*p* < 0.05) than the control sample. The cohesiveness of the samples decreased with the increased level of GBF additions. The GBF addition also influenced the resilience parameter. The data evolution of this parameter corresponded to that of the elasticity parameter that was previously determined.

#### 3.4.4. Crumb Microstructure of Breads Samples

The microstructure of the breadcrumbs was characterized by pores that are formed due to the release of gas during the fermentation process of the dough and gas retention during fermentation and bread baking [56]. A bread of a good quality is characterized by small and uniform pores, which is desirable for a good porosity [57].

From Figure 4, it may be noticed that the addition of germinated bean flour in the bread-making recipe led to an increased in the average pore diameter and, at the same time, to a decreased in pore density. With the increased level of GBF addition, the pores size increased more and more, and their density decreased. These data are similar with those obtained by Protonotariou et al. [58].

#### 3.4.5. Sensory Analysis of Breads Samples

The sensory analysis highlights consumer preferences. Performing a sensory analysis tests is necessary to remove the risk of market failure, if the products are to be placed on the market [59]. Appearance, color, taste, smell, texture, aroma and overall acceptability showed how the quality of the bread products was assessed.

From Table 7, it may be noticed that an addition of 5%, 10% and 15% GBF in wheat flour had the effect of improving the sensory parameters compared to the control sample without any additions. Levels higher than a 20% GBF addition in wheat flour ended in bread samples less appreciated by consumers compared to the control. From a sensory point of view, the bread samples with 5% and 10% GBF were best appreciated by the panelists.

#### 3.4.6. Effect of GBF Addition on Bread Compositional Analysis

The bread compositional analysis shown in Table 8 was made in order to provide some nutritional data related to the macronutrient composition of the bread obtained with different levels of GBF additions in the wheat flour. The most significant differences (*p* < 0.05) in the nutritional profile were for the protein, ash and carbohydrates contents because of the GBF composition, which presented higher protein and ash contents than the wheat flour, which it partially substituted.

## 4. Discussion

### 4.1. Dough Rheological Properties

#### 4.1.1. Dough Rheological Properties during Mixing and Extension

According to Consistograph data, the GBF addition decreased the water absorption value of the mixes up to 51.4%. This may be due to the fact that, during the germination process, a part of the starch and protein contents of the bean grains were lost. Additionally, these components changed their structures, which may have had a decreasing effect on the water absorption value. During germination, the protein part may be hydrolyzed by proteases into amino acids and lower molecular peptides, and starch may be hydrolyzed by amylases to dextrins [60]. The tolerance to mixing increased with increasing the GBF level from 0% to 15%, indicating an increase of the gluten network stability that was not expected. However, these results were in agreement with different authors data that reported an increase of the dough stability with an increase of the different levels of germinated legumes addition in wheat flour. Sadowska et al. [61] reported an increase of dough stability after pea germination and a decrease of its value when raw peas were incorporated into a dough recipe. Rosales-Juárez et al. [62] reported an increase of dough stability with the increase level of germinated soybean flour addition, whereas Razaviey al. [63] reported a constant stability value by a germinated lentil flour addition in wheat flour. Samples with the highest amounts of GBF addition (GBF_20 and GBF_25) presented a decreased tolerance to the mixing values, probably due to the increase level of α-amylase content from the mix flour, which exerted a hydrolytic effect on the starch, leading to degradation and to an increased maltose level. Additionally, by the increase of the GBF concentration of wheat flour, the gluten content from the flour mixes decreased, and consequently, the dough lost its stability. The dough consistency after 250 s and 450 s was reduced with the increased level of GBF addition. This behavior may be related to a change in the dough viscosity through GBF addition. GBF contains a high amount of protein, diet fiber and carbohydrates. The main components of diet fiber from bean grains are pentosans, pectins, hemicellulose, lignin and cellulose [36].These components may interfere with gluten during mixing, increasing the viscosity and leading to lower D250 and D450 values. A decrease of dough consistency during mixing was also reported by Kohajdová et al. [64] for dough samples in which beans and lentils were incorporated into wheat flour.

To Alveograph, the increased of maximum pressure is up to 18.62% for the sample with the highest level of GBF addition in wheat flour. This effect may be due to the increase of viscosity of dough samples with GBF addition, which lead to a denser dough with a higher resistance. The data obtained suggested that the chemical action of GBF on dough system was less than the physical one. Even if GBF is a germinated flour which is an active one from the enzymatic point of view its enzymatic effects on dough rheological properties on Alveograph are not evident ones. The maximum pressure of wheat flour dough is higher for all samples with GBF addition compared to the control one. This may be caused by the interactions between gluten proteins and GBF fibers, which values ranged between 14% and 19% on raw bean [36]. The presence of a high level of fiber in GBF content may interfere with wheat flour proteins during mixing causing an increase of dough resistance to deformation or maximum pressure [65]. Dough extensibility and index of swelling indicators of the dough handling characteristics were reduced by GBF addition. According to Mohammed et al. [66], the decrease of extensibility may be caused by the increasing of sulfhydryl groups (SH) or thiol group’s level, which may oxidize the dough with oxygen through the mechanical action. The conversion of SH-bonds in SS-bonds may lead to gluten and dough solidification (decreased extensibility). A decrease of dough extensibility has also been reported by Hallén et al. [67] when high levels of germinated cowpea flour were incorporated in dough recipe whereas Rosales-Juárez et al. [62] reported that germinated soybean flour did not affected this parameter. The resulting effect on extensibility and maximum pressure values becomes evident in the configuration ratio of the Alveograph curve. The addition of GBF increased the P/L ratio (3.46 for sample with 25% GBF addition versus 1.38 in the control).The decrease of baking strength, which is a predictor of dough strength, indicates its weakening by GBF addition. This may be due to the gluten dilution effect induced by GBF addition which does not contain gluten.

#### 4.1.2. Dough Rheological Properties during Fermentation and Falling Number Values

During the fermentation process, the maximum height of gaseous production (H’m) increased when low levels of GBF were incorporated in dough recipe and decreased when high levels of GBF were added in wheat flour. This may be due to the increase level of α-amylase activity through GBF addition in wheat flour. According to our data, the falling number value decreased up to 25.28% for GBF_25 sample compared to the control one. The FN value expressed α-amylase activity in mixes through their viscosity, which is a much lower one, as the α-amylase enzyme activity from the mixes increased [50]. The H’m increased up to 23.11% for the GBF_10 sample compared to the control one, maintaining high-level values for all samples except the GBF_25 one. These H’m high values are due to the gases formed in dough system during fermentation process. As α-amylase activity increases in the dough system, the starch degradation process intensifies, leading to a more maltose level that may be used by the yeast [68]. As a result, the gases gradually increased, leading to a stretched of the three-dimensional structure of dough system. As a consequence, the H’m increased up to a level of dough ability to retain gases formed during the fermentation process. A decreased of H’m values to high levels of GBF addition in wheat flour was due to gluten network weakening as an effect of it dilution through GBF addition in wheat flour. Even if the gas production continues to take place in the dough system, its ability to retain gases decreased and the lost gases increased [69]. This may be noticed through VT and VR values, which presented higher values for all samples with GBF addition in wheat flour compared to the control one, except for the GBF_25 sample for which the lowest VT and VR values were recorded.

#### 4.1.3. Dough Fundamental Rheological Properties

The effect of GBF addition in wheat flour on dynamic dough rheological properties shown that tan *δ* increased in a frequency-dependent manner. *G’* and *G”* increased with high levels of GBF addition in wheat flour, indicating that an addition GBF improved the dough viscoelasticity. The tan *δ* lower than 1 for all dough samples showed a solid-like behavior of the dough samples [70]. Tan *δ* presented higher values for dough samples with GBF addition, which indicated the decreased in the ratio of elastic structure. The effects of temperature on dough rheological properties with different levels of GBF addition in wheat flour were shown on temperature sweep graphs. As it may be seen, the paste temperature of the control sample was lower than the dough samples with a GBF addition in dough recipe. The increased values of temperature led to an increased value of dough viscoelasticity due to the starch gelatinization process. With the increasing degree of starch gelatinization, the dough viscoelasticity decreased. The starch gelatinization depends on α-amylase activity increased [50]. It is well-known that, through the germination process, the α-amylase activity from beans increased, a fact highlighted by the falling number value of the wheat GBF mixes, which decreased with the increased level of GBF addition in wheat flour. Under the α-amylase effect, the starch degradation was intensified, which led to a higher gelatinization capacity of mix flour and, therefore, to a decrease of it viscoelastic moduli [71]. Therefore, the samples with a GBF addition, which represented a higher level of α-amylase in the dough system, will present lower *G’* and *G”* values and higher tan *δ* values compared to the control sample when dough mixes are in the starch gelatinization phase. In the cooking stability phase, when α-amylase acted on starch due to the fact that it was in the optimum activity range, the dough viscoelasticity decreased for all the dough samples, even more with the increased level of the α-amylase from the mixed flours [49]. Therefore, the samples with a GBF addition will present lower *G’* and *G”* values and higher tan *δ* values compared to the control one in the cooking stability phase.

### 4.2. Dough Microstructure

The microstructure of the control sample (Figure 3A–F) showed a clearly defined dough network consisting in starch granules and inter-dispersed protein (gluten), which is squeezed into the starch matrix. The images obtained for the dough samples with different levels of germinated bean flour addition shows significant difference among the samples. The spatial distribution of protein and starch within the dough structure is changing with the increase level of GBF addition, as it may be seen from Figure 3A–F. The GBF addition led to a higher red area and a less green one, indicating a more protein content and a lower starch one in the dough network. This fact is explicable, taking into account that GBF contain a double amount of proteins compared to the wheat flour. When high levels of GBF were incorporated into the dough, the starch granules appeared more dispersed within the protein phase. Contrary, for the dough samples with low levels of GBF additions, the starch granules clumped together in the dough network, being more concentrated into large areas, and the protein phase was thicker.

### 4.3. Bread Quality Evaluation

#### 4.3.1. Bread Physical Characteristics

Usually, the addition of legume flours in the bread-making recipe has the effect of negatively influencing the physical characteristics of the finished products [64], but germination leads to the attenuation of this influence. The addition of germinated bean flour had the effect of influencing the value of the specific volume, porosity and elasticity of the bread samples. The influence of GBF addition on the specific volume can be explained by the changes in the process of starch gelatinization and protein aggregation. The fact that the addition did not significantly reduce the specific volume of the bread, not even at a high level of GBF addition in wheat flour (25%), showed that the formation and stabilization of the gas network during the dough proofing and baking were not very much affected by the GBF addition. The increase in the specific volume of bread could also be explained by the fact that, during the germination process, the solubility of proteins increased, which led to the improvement of the foaming and emulsifying activity of bean flour [6,72,73]. During germination, the amount of fermentable sugars increased, along with the activation of amylase, which facilitated the activity of yeast during the fermentation of the dough, which resulted in the formation of a larger amount of CO_2_. This could explain the increase in bread volume with the addition of GBF in wheat flour [7,74]. Probably, the increased value of a specific volume of bread sample was due to the increased level of α-amylase by GBF addition (activated through the germination process of bean flour) [75]. A decrease in the specific volume of the bread after exceeding the level of 15% GBF addition could be explained by the fact that there was a decrease in the amount of starch in the dough matrix, which had the effect of decreasing the water absorption capacity and emulsification capacity, as explained by Benítez et al. in a previous study [76]. The decrease in a specific volume with the same level of increase of the GBF addition in wheat flour is also explained by the fact that the amount of gluten proteins decreases, and the proteins in the bean composition compete with wheat proteins to absorb water while mixing the ingredients [77]. Therefore, the decrease in bread specific volume may be attributed to three concomitant factors: the dilution of gluten, interference of GBF proteins with the formation of gluten matrix and changes in the enzymatic activity, which has the effect of influencing the properties of starch. According to the data obtained, it can be concluded that an addition of up to 15% GBF leads to a bread with a good specific volume, because gluten proteins are still able to aggregate and to retain gases formed during the dough fermentation [78]. The results obtained are in agreement with different studies on germinated legume flour additions in wheat flour, which reported the fact that an addition of 10–15% germinated legume flour does not negatively affect the volumes of bread samples [6,79].

A higher porosity of the bread is related to an increased volume [52], which is also highlighted in this study. Studies show that the porosity of bread actually gives it its spongy character, its softness, which is an indicator of quality for consumers [80]. The decrease in the porosity value is explained by the fact that the replacement of wheat flour with germinated bean flour led to a decrease in the amount of gluten, which led to weaker dough, which negatively influenced the stage of bread leavening and its pore formation [81,82]. Additionally, the decrease in porosity may be caused by a gluten matrix disruption [57]. The increase in porosity is explained by the fact that the GBF addition increased gas production during the dough fermentation [83] and led to the appearance of pores much denser. The decrease in porosity to an addition of more than 20% GBF in wheat flour can be also due to an increase in the amount of fibers [84] from the GBF content. Beans contain a much higher amount of fiber than white wheat flour [25,85,86], which represents 14–19% of the seed’s weight [36]. The increase in elasticity value can be attributed to the improvement of the gluten structure [87]. Improving the elasticity of the bread is desirable, because it is an indicator of the freshness and quality of the bread. The improvement in the elasticity of the bread samples may be due to the α-amylase, which was activated during the bean germination process [88].

#### 4.3.2. Color Analysis of Breads Samples

Regarding the color parameters values, studies have shown that the increase of the value of the parameter *a** (the value of red) can be attributed to the pigment present in the outer shell of the borlotti-type beans. The decrease in the brightness of the samples can be attributed to the increase in the amount of protein [89]. Different studies reported that beans contain a higher amount of protein than white wheat flour [23,90]. Taking into account that germination leads to an increase in the amount of protein in beans [91], then it may be concluded that the samples of bread with the GBF addition will contain a higher amount of proteins. The darkening of the bread samples with the increase amount of GBF addition could also be explained by the fact that the amount of phenolic compound increases [92]. Different studies have been reported that the germination process leads to an increase in the amount of phenolic compounds in the grains [39]. Phenolic compounds give a darker coloration to flour and, implicitly, to the finished products [93]. The increase in the value of the *b** indicator (yellow color) is also related to the increase in the amount of protein with the increase level of GBF addition in wheat flour [54]. A higher amount of protein favors the Maillard reaction, which has a direct influence on the color of the bread samples [94]. However, the color of the bread samples is mainly influenced by the raw materials used and their proportion [95,96] but, also, by the reactions that take place during the baking process, especially on bread crust. Other studies have correlated the dense structure of bread pores with the decrease of brightness and the increase of red and yellow values [58]. Therefore, correlating the results obtained with the colorimeter device and the images obtained with the stereomicroscope, it can be concluded that the results obtained in this study are in accordance with other ones.

#### 4.3.3. Texture Profile Analysis of Breads Samples

The firmness of the bread was influenced by GBF addition in wheat flour. This may be influenced by the interaction between protein and starch, which changed due to GBF addition in wheat flour, a fact which may lead to an increase in the firmness parameter [97]. The influence of the values of the gumminess parameter is due to the modification of the gluten networks structure due to the GBF addition [98]. Chewiness is a dimensionless size and characterizes the energy needed to chew the food and is characterized by the parameters: firmness, cohesiveness and springiness [99]. The reduction of the chewiness value could be attributed to the weakness in the starch–gluten network. This weakness was associated with swelling of starch granules and water absorption of gluten network during dough formation [100]. Resilience characterizes the force and speed involved in the recovery of a food when a deforming force is removed [101]. In this study, this parameter was improved due to the addition, not exceeding 20%.

#### 4.3.4. Crumb Microstructure of Breads Samples

Regarding the crumb microstructure, data have shown that the GBF addition has a direct influence on the size of the pores of the bread and their density. Thus, particles of different sizes of GBF addition absorb water differently, which has an influence on the crumb microstructure. Thus, it was shown that larger particles lead to a loaf of bread with a more compact structure, with smaller pores and thicker, as it may be seen in the present study [102]. The change in the microstructure of the bread crumb is also due to the influence of the GBF addition on the gluten matrix, which led to the limitation of dough expansion during fermentation and difficulty in gas retention, which had the effect of increasing pore density and reducing their size [103].

#### 4.3.5. Sensory Analysis of the Bread Samples

Regarding the results of the sensory analysis obtained, it can be concluded that an addition of 5%, 10% and 15% germinated bean flour, respectively, led to a better assessment by the panelists compared to the sample without any GBF addition due to the nutritional composition of the beans but, also, due to the advantages of the germination process over the sensory profile of the bean grains. Studies shown that, during the germination process, take place the activation of enzymes in the bean grains [104], enzymes that will have an important role in obtaining a quality bread, with an improved appearance, texture and specific volume [105,106]. Studies also shown that the germination process has an influence on taste because during germination, reducing sugars and amino acids are released from the grains, which reacts later, during the baking process, forming specific products of the Maillard reaction (which influences the taste and color of the finished products) [107]. Additionally, the activation of endogenous amylolytic enzymes during germination results in the transformation of starch into oligosaccharides and sugars, which gives to the germinated bean flour a certain sweetness and, implicitly, to the food products in which it is incorporated—the bread, in this case. Studies have also shown that germination promotes the formation of caramel-smelling compounds [108]. Thus, it can be concluded that in this study the sensory perception of consumers related to bread with a GBF addition of maximum 15% level in wheat flour was not negatively affected. Additionally, it can be concluded that GBF could be successfully used to improve wheat bread from the nutritional point of view. Similar results were obtained for bread samples in which bean flour were incorporated in bread recipe [109], but if we take into account the benefits of the germination process, then the addition of GBF in wheat flour is a more desirable one in bread making. Thus, it can be concluded that the improvement of the sensory profile of bread with the addition of germinated bean flour is mainly due to the germination process, which improves the profile of the grains subjected to this process and implicitly to the bread in which the flour from these grains was incorporated. Similar results regarding the improvement of the sensory profile of bread by adding germinated legumes in wheat flour were obtained in other previous studies [110].

#### 4.3.6. Compositional Analysis of Bread Samples

All the macronutrient content of the bread samples were significant different (*p* < 0.05) compared to the control sample. The protein content significant increased (*p* < 0.05) with the increase level of GBF addition in wheat flour up to 44.2% for the GBF_25 sample. This increased are due to the high protein content of the GBF whose protein content is twice as high as that of wheat flour sample. A significant increased (*p* < 0.05) of fat and ash content also may be seen for the bread samples with different levels of GBF additions up to 23.45% and 80.39% for the GBF_25 sample. This increase is due to the high level of fat and ash from GBF sample compared to the wheat flour, which presented lower values for these parameters. The carbohydrates content significant (*p* < 0.05) decreased with the increase level of GBF addition in wheat flour due of the substitution of wheat flour (which is a refined one with high starch content) by GBF with a high protein content compared to wheat flour. The caloric values of the bread samples with GBF addition were lower (*p* < 0.05) than the control sample. However, the decreases were a small one of only 0.49% for the GBF_20 sample, for which was recorded the lowest caloric value.

## 5. Conclusions

The incorporation of germinated bean flour (as a partial substitution of wheat flour) in wheat flour led to significant changes on the dough rheological properties, microstructural properties and bread quality. The presence of active enzyme, dietary fiber and non-gluten proteins from GBF were responsible for the decrease of the dough consistency, baking strength and extensibility mainly due to the gluten network modifications shown by EFLM. The dilution effect of the gluten network would be responsible for the less-elastic structure highlighted by a tan δ value increase. A more α-amylase activity in the dough samples due to a GBF addition led to a decrease in the dynamic moduli values when the dough temperature increased, showing a less viscoelasticity fact highlighted also by a falling number values decrease. During fermentation, the maximum height of gaseous production and total CO_2_ volume production were improved up to a certain level of GBF addition due also to an increase of the α-amylase activity. However, at high levels of GBF addition, these values decreased, probably due to the gluten dilution from the dough system. For the bread quality, all the analyzed parameters were improved up to a 15% GBF addition level in wheat flour. Regarding the color parameters values, our data shown a decrease of *L** and an increase of the *a** and *b** values for the bread crumbs and crust, indicating a darkening of their color. The crumb microstructure of the bread samples showed larger pore sizes at high levels of GBF addition due to the GBF dilution effect on the gluten matrix. From a sensory point of view, the bread sample characteristics were well-appreciated up to a 15% GBF addition level in wheat flour.

## Figures and Tables

**Figure 1 foods-10-01542-f001:**
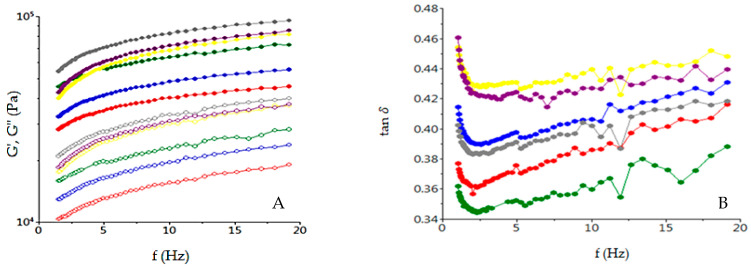
Evaluation with frequency of *G’*(open symbols), *G”*(solid symbols) (**A**) and tan *δ* (**B**) for the samples with different levels (-●-0%; -●-5%; -●-10%; -●-15%; -●-20%; -●-25%) of germinated bean flour additions.

**Figure 2 foods-10-01542-f002:**
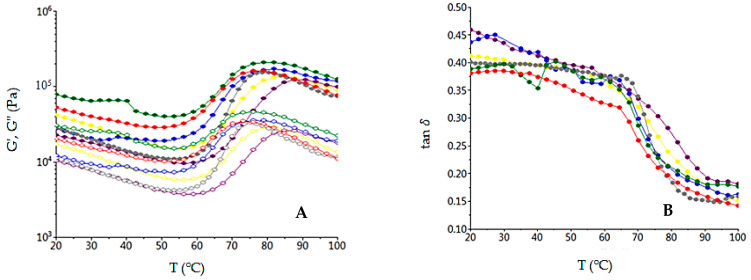
Evaluation with temperature of the G′ values (open symbols), G′′ (solid symbols) (**A**) and tan *δ* (**B**) for the samples with different levels (-●-0%; -●-5%; -●-10%; -●-15%; -●-20%; -●-25%) of germinated bean flour additions.

**Figure 3 foods-10-01542-f003:**
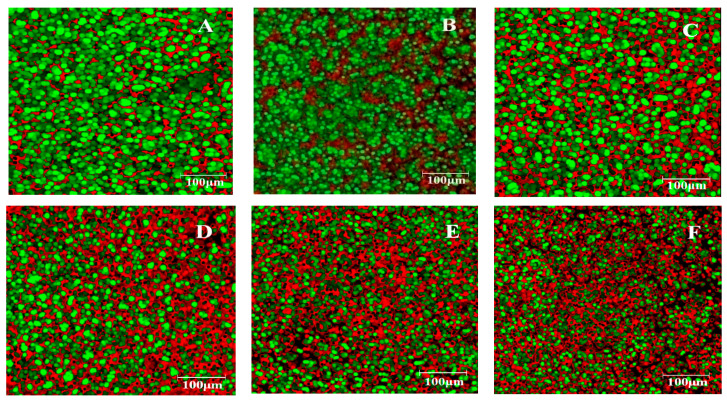
Microstructure taken by EFLM of the wheat dough with germinated bean flour (GBF) at different levels: 0% (**A**), 5% (**B**), 10% (**C**), 15% (**D**), 20% (**E**) and 25% (**F**). Red, protein; Green, starch granules.

**Figure 4 foods-10-01542-f004:**
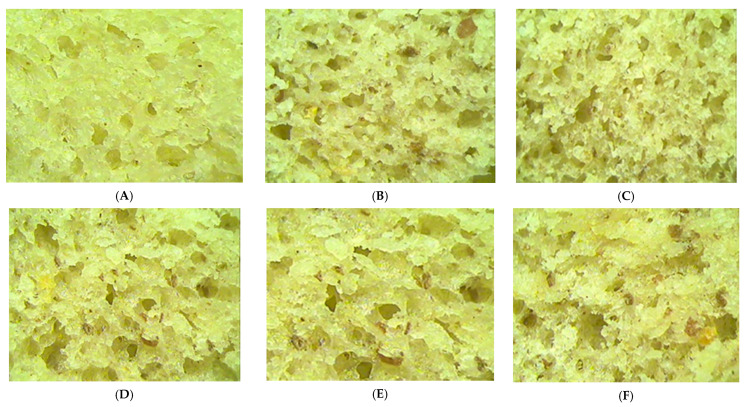
Microstructure of the crumb samples with germinated bean flour (GBF) at different levels: 0% (**A**), 5% (**B**), 10% (**C**), 15% (**D**), 20% (**E**) and 25% (**F**).

**Table 1 foods-10-01542-t001:** Consistograph parameters of the dough samples with different levels of germinated bean flour (BGF) additions.

Dough Samples	WA (%)	Tol (s)	D250 (mb)	D450 (mb)
Control	54.3 ± 0.10 ^c^	214 ± 1.00 ^a^	394 ± 2.00 ^f^	943 ± 1.00 ^f^
GBF_5	53.7 ± 0.15 ^c^	229 ± 1.52 ^a^	270 ± 0.57 ^e^	821 ± 3.05 ^e^
GBF_10	53.0 ± 0.10 ^b^	246 ± 2.51 ^c^	231 ± 1.00 ^d^	808 ± 2.00 ^d^
GBF_15	52.7 ± 0.05 ^b^	254 ± 1.52 ^d^	216 ± 1.52 ^c^	776 ± 2.51 ^c^
GBF_20	52.0 ± 0.05 ^a^	238 ± 1.00 ^c^	183 ± 1.52 ^b^	760 ± 2.08 ^b^
BGF_25	51.5 ± 0.15 ^a^	226 ± 1.52 ^b^	150 ± 2.00 ^a^	617 ± 2.08 ^a^

WA, water absorption; Tol, tolerance to mixing; D250, dough consistency after 250 s; D450, dough consistency after 450 s. The results are the mean ± standard deviation (*n* = 3). Dough samples containg germinated bean flour, GBF: a–f, mean values in the same column followed by different letters are significantly different (*p* < 0.05).

**Table 2 foods-10-01542-t002:** Alveograph parameters of the dough samples with different levels of germinated bean flour (BGF) additions.

Dough Samples	P (mm)	L (mm)	G (mm)	W (10^−4^ J)	P/L
Control	104 ± 2.51 ^a^	72 ± 1.15 ^c^	19.4±0.28 ^d^	301 ± 5.13 ^d^	1.43 ± 0.05 ^a^
GBF_5	119 ± 1.15 ^b^	63 ± 4.72 ^bc^	18.1 ± 0.30 ^cd^	276 ± 6.42 ^d^	1.88 ± 0.15 ^b^
GBF_10	114 ± 1.15 ^b^	51 ± 2.08 ^ab^	16.2 ± 0.60 ^bc^	220 ± 5.85 ^c^	2.35 ± 0.25 ^c^
GBF_15	118 ± 1.15 ^b^	48 ± 2.08 ^ab^	15.4 ± 0.37 ^ab^	203 ± 5.13 ^bc^	2.43 ± 0.12 ^c^
GBF_20	121 ± 1.52 ^b^	41 ± 2.88 ^a^	14.3 ± 0.46 ^ab^	187 ± 6.24 ^ab^	2.84 ± 0.15 ^d^
GBF_25	123 ± 2.08 ^b^	33 ± 1.73 ^a^	13.6 ± 0.40 ^a^	159 ± 4.04 ^a^	3.74 ± 0.24 ^e^

P, maximum pressure; L, dough extensibility; G, index of swelling; W, baking strength; P/L, configuration ratio of the Alveograph curve. The results are the mean ± standard deviation (*n* = 3). Dough samples containg germinated bean flour, GBF: a–e, mean values in the same column followed by different letters are significantly different (*p* < 0.05).

**Table 3 foods-10-01542-t003:** Rheofermentometer parameters and falling number values of dough samples with different levels of germinated bean flour (BGF) additions.

Dough Samples	H’m (mm)	VT (mL)	VR (mL)	CR (%)	FN (s)
Control	65.9 ± 0.30 ^a^	1532 ± 2.51 ^b^	1228 ± 2.51 ^b^	80.1 ± 0.50 ^b^	350 ± 3.29 ^d^
GBF_5	70.9 ± 0.85 ^ab^	1644 ± 5.85 ^c^	1289 ± 2.00 ^d^	78.4 ± 0.42 ^ab^	331 ± 2.51 ^c^
GBF_10	80.4 ± 1.21 ^b^	1951 ± 2.51 ^f^	1363 ± 3.6 ^e^	69.8 ± 0.75 ^a^	282 ± 2.04 ^b^
GBF_15	72.6 ± 1.00 ^ab^	1679 ± 2.51 ^e^	1259 ± 2.64 ^c^	74.9 ± 0.37 ^ab^	278 ± 2.00 ^ab^
GBF_20	69.6 ± 1.56 ^a^	1650 ± 2.51 ^d^	1238 ± 2.51 ^bc^	75.0 ± 1.01 ^ab^	270 ± 1.52 ^ab^
GBF_25	64.8 ± 3.81 ^a^	1392 ± 3.05 ^a^	1146 ± 1.00 ^a^	82.3 ± 0.62 ^b^	262 ± 3.05 ^a^

H’m, maximum height of gaseous production; VT, total CO_2_ volume production; VR, volume of the gas retained in the dough at the end of the test; CR, retention coefficient; FN, falling number value. The results are the mean ± standard deviation (*n* = 3). Dough samples containg germinated bean flour, GBF: a–e, mean values in the same column followed by different letters are significantly different (*p* < 0.05).

**Table 4 foods-10-01542-t004:** Physical characteristics of the bread samples with different levels of germinated bean flour (GBF) additions.

Bread Samples	Specific Volume (cm^3^/100 g)	Porosity (%)	Elasticity (%)
Control	331.5 ± 0.74 ^c^	67.4 ± 0.86 ^b^	91.3 ± 0.57 ^c^
GBF_5	352.4 ± 0.75 ^d^	70.6 ± 0.36 ^c^	91.3 ± 1.00 ^c^
GBF_10	359.2 ± 0.75 ^de^	73.7 ± 0.50 ^d^	92.3 ± 0.57 ^c^
GBF_15	367.2 ± 2.15 ^e^	72.4 ± 0.77 ^cd^	91.66 ± 1.15 ^c^
GBF_20	312.9 ± 2.27 ^b^	68.4 ± 0.80 ^b^	83.66 ± 1.15 ^b^
GBF_25	292.4 ± 7.94 ^a^	59.1 ± 0.70 ^a^	75.3 ± 0.57 ^a^

The results are the mean ± standard deviation (*n* = 3). Bread samples containg germinated bean flour, GBF: a–e, mean values in the same column followed by different letters are significantly different (*p* < 0.05).

**Table 5 foods-10-01542-t005:** Color parameters of the bread samples with different levels of germinated bean flour (GBF) additions.

Bread Samples	Crust Color			Crumb Color		
	*L**	*a**	*b**	*L**	*a**	*b**
Control	76.25 ± 0.94 ^e^	3.44 ± 0.27 ^a^	3.14 ± 0.43 ^a^	66.37 ± 0.88 ^d^	−4.62 ± 0.32 ^a^	1.69 ± 0.22 ^a^
GBF_5	67.48 ± 1.11 ^d^	6.22 ± 0.64 ^b^	4.81 ± 0.44 ^b^	64.66 ± 0.54 ^cd^	−3.70 ± 0.51 ^a^	2.49 ± 0.36 ^ab^
GBF_10	58.49 ± 1.65 ^c^	7.90 ± 0.41 ^b^	5.21 ± 0.43 ^b^	62.03 ± 0.50 ^bc^	−2.63 ± 0.48 ^b^	3.31 ± 0.34 ^bc^
GBF_15	53.23 ± 1.02 ^b^	13.21 ± 0.90 ^c^	5.57 ± 0.36 ^bc^	60.11 ± 0.23 ^ab^	−1.86 ± 0.25 ^b^	3.82 ± 0.41 ^cd^
GBF_20	44.47 ± 1.77 ^a^	14.15 ± 0.33 ^c^	6.56 ± 0.33 ^c^	59.37 ± 0.76 ^ab^	−0.61 ± 0.12 ^c^	4.37 ± 0.42 ^de^
GBF_25	43.80 ± 0.37 ^a^	15.71 ± 0.95 ^d^	8.83 ± 0.53 ^d^	57.54 ± 0.39 ^a^	0.38 ± 0.08 ^d^	4.51 ± 0.62 ^e^

The results are the mean ± standard deviation (*n* = 10). Bread samples containg germinated bean flour, GBF: a–e, mean values in the same column followed by different letters are significantly different (*p* < 0.05).

**Table 6 foods-10-01542-t006:** Texture parameters of the bread samples with different levels of germinated bean flour (GBF) additions.

Bread Samples	Firmness (*N*)	Gumminess (*N*)	Chewiness (*J*)	Cohesiveness (Adimensional)	Resilience (Adimensional)
Control	9.01 ± 3.06 ^a^	7.23 ± 1.73 ^c^	7.23 ± 1.73 ^d^	0.82 ± 0.03 ^d^	1.72 ± 0.04 ^d^
GBF_5	8.75 ± 4.04 ^a^	7.05 ± 1.57 ^c^	7.05 ± 1.57 ^c^	0.80 ± 0.02 ^cd^	1.46 ± 0.06 ^c^
GBF_10	8.63 ± 1.72 ^a^	6.22 ± 3.05 ^b^	6.22 ± 2.05 ^b^	0.72 ± 0.02 ^bc^	1.40 ± 0.05 ^bc^
GBF_15	8.46 ± 3.88 ^a^	5.86 ± 4.10 ^a^	5.86 ± 3.10 ^a^	0.71 ± 0.02 ^bc^	1.26 ± 0.11 ^b^
GBF_20	10.26 ± 4.72 ^b^	7.93 ± 3.73 ^d^	7.93 ± 3.73 ^e^	0.65 ± 0.05 ^b^	1.06 ± 0.04 ^a^
GBF_25	17.72 ± 3.71 ^c^	8.25 ± 2.47 ^e^	8.25 ± 4.47 ^f^	0.52 ± 0.05 ^a^	1.02 ± 0.03 ^a^

The results are the mean ± standard deviation (*n* = 3). Bread samples containg germinated bean flour, GBF: a–f, mean values in the same column followed by different letters are significantly different (*p* < 0.05).

**Table 7 foods-10-01542-t007:** Sensory analysis of the bread samples with different levels of germinated bean flour (GBF) additions.

Bread Samples	Appearance	Color	Taste	Smell	Texture	Flavor	Global Acceptability
Control	7.4 ± 0.15 ^c^	8.0 ± 0.15 ^b^	7.8 ± 0.21 ^b^	7.7 ± 0.91 ^ab^	7.7 ± 0.21 ^bc^	7.3 ± 0.61 ^b^	7.7 ± 0.21 ^b^
GBF_5	8.6 ± 0.32 ^d^	8.4 ± 0.31 ^b^	8.4 ± 0.26 ^b^	8.4 ± 0.45 ^b^	8.6 ± 0.65 ^c^	8.3 ± 0.61 ^b^	8.5 ± 0.32 ^c^
GBF_10	8.7 ± 0.36 ^d^	8.2 ± 0.71 ^b^	8.1 ± 0.15 ^b^	8.2 ± 0.31 ^b^	8.3 ± 0.31 ^bc^	8.2 ± 0.76 ^b^	8.4 ± 0.23 ^c^
GBF_15	7.9 ± 0.25 ^c^	8.0 ± 0.21 ^b^	8.0 ± 0.40 ^b^	7.4 ± 0.25 ^ab^	7.2 ± 0.35 ^b^	7.4 ± 0.10 ^b^	7.5 ± 0.40 ^b^
GBF_20	6.7 ± 0.15 ^b^	6.5 ± 0.2 ^a^	5.8 ± 0.10 ^a^	8.0 ± 0.5 ^b^	5.8 ± 0.50 ^a^	6.0 ± 0.25 ^a^	6.1 ± 0.31 ^a^
GBF_25	5.2 ± 0.25 ^a^	5.7 ± 0.21 ^a^	5.2 ± 0.15 ^a^	6.5 ± 0.26 ^a^	5.1 ± 0.10 ^a^	5.7 ± 0.11 ^a^	5.7 ± 0.21 ^a^

The results are the mean ± standard deviation (*n* = 20). Bread samples containg germinated bean flour, GBF: a–d, mean values in the same column followed by different letters are significantly different (*p* < 0.05).

**Table 8 foods-10-01542-t008:** Compositional analysis of the bread samples with different levels of germinated bean flour (GBF) additions.

Bread Samples	Moisture (%)	Protein (%)	Fat (%)	Ash (%)	Carbohydrates (%)	Energy (kcal/100 g)
Control	44.72 ± 0.02 ^b^	8.80 ± 0.01 ^a^	0.81 ± 0.01 ^a^	0.51 ± 0.01 ^a^	45.14 ± 0.04 ^f^	223.13 ± 0.14 ^e^
GBF_5	44.90 ± 0.30 ^e^	9.22 ± 0.01 ^b^	0.86 ± 0.01 ^b^	0.56 ± 0.01 ^b^	44.45 ± 0.01 ^e^	222.41 ± 0.08 ^bc^
GBF_10	44.79 ± 0.01 ^c^	10.41 ± 0.01 ^c^	0.89 ± 0.00 ^c^	0.67 ± 0.01 ^c^	43.21 ± 0.04 ^d^	222.59 ± 0.07 ^cd^
GBF_15	44.83 ± 0.01 ^cd^	11.23 ± 0.01 ^d^	0.91 ± 0.00 ^c^	0.76 ± 0.00 ^d^	42.24 ± 0.02 ^c^	222.20 ± 0.13 ^ab^
GBF_20	44.88 ± 0.02 ^de^	11.91 ± 0.01 ^e^	0.97 ± 0.01 ^d^	0.83 ± 0.01 ^e^	41.40 ± 0.05 ^b^	222.04 ± 0.07 ^a^
GBF_25	44.65 ± 0.02 ^a^	12.69 ± 0.01 ^f^	1.00 ± 0.00 ^e^	0.92 ± 0.01 ^f^	40.73 ± 0.02 ^a^	222.75 ± 0.08 ^d^

The results are the mean ± standard deviation (*n* = 3). Bread samples containg germinated bean flour, GBF: a–f, mean values in the same column followed by different letters are significantly different (*p* < 0.05).

## Data Availability

Not applicable.

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
