# Peer review of "Dough Rheological Properties, Microstructure and Bread Quality of Wheat-Germinated Bean Composite Flour"

_foods, 2021, doi:10.3390/foods10071542_

Round 1

Reviewer 1 Report

Article Type: Article
Title: Dough rheological properties, microstructure and bread quality of wheat-germinated bean composite flour

Journal: Foods-1253511

 It is important to know the impact of adding healthy ingredients in the dough properties and technological aptitude of baked goods.

The article contains new information, the aim of the work is clearly established, discussion and conclusions are well documented. Besides technological and sensory impact of adding germinated bean flour to wheat flour, it would be very interesting to evaluate the impact in terms of nutrition and bioactivity.

However, my big concern is related to sample size and statistical analysis. All data were made in duplicate and one-way analysis of variance (ANOVA) with Tukey’s test was used to determine the significant differences at 5% level. To apply Tukey test, one of the assumptions is that the groups associated with each mean in the test are normally distributed and other is that there is homogeneity of variance. In theory, sample size is not important for Tukey test, but power of these results is dependent on sample size. In my opinion, a sample size of two is usually unpublishable, since it increases the risk that obtained results are not valid. In addition, some statistical results obtained for several parameters are strange (letters obtained for loaf volume and porosity, for example).

Abstract

    1. Why the Authors use the abbreviation BGF for Germinated bean flour, instead of GBF?
    2. Please use L* instead of L.

Keywords: “germinated bean flour” instead of “germinated bean”.

  1. The entire document must be revised due to inaccuracies in some words and the absence of spaces between words and between words and bibliographic references.

 Introduction

    1. Lines 36-69: Authors say, “Through germination, the enzymatic activity of grains are increased which has the effect of facilitating the digestion of compounds such as starch and proteins and therefore makes germinated legumes to be used successfully in food where enzyme activity is required”. It is of great value to explain the technological importance of enzymes, especially amylases, on the dough fermentation process.
    2. Lines 49 and 61: Please avoid “etc.”.
    3. Lines 49/52: About beans protein, something should be said about its aminoacids profile.
    4. Lines 52/54: Authors say, “Beans also presents a high content of bioactive substances such as tannins, flavonoids, phenolic compounds and other antioxidants”. Tannins and flavonoids are phenolic compounds… Please use a reference to support this sentence.
    5. Line 67: “This process increases the bioavailability of nutrients, such as amino acids like lysine which is deficient in wheat flour [32], minerals present in the beans such as magnesium, iron, calcium, vitamins in especially B complex 65 in which beans are rich [33], decreases the amount of antinutrient compounds, such as phytic acid [34], trypsin inhibitors [32], and activates hydrolytic enzymes in the grain [35]”. Please add what is in bold.

Materials and Methods

  1. The type of flour should be indicated since it has high impact on the rheology and technological properties of wheat flour. Is it type 650 (line 144)?
  2. Were doughs/breads prepared in duplicate? This is not specified… If not, the standard error only represents analytical measurement errors, and weakens your statistical power for identifying significant differences.
  3. Number of replicates should be presented in all experimental sections.
  4. Consistograph: Tests were made using an adapted hydration level to achieve 2200 ± 100 mm H2O? Tests were carried out at 14% or 15% moisture basis?
  5. Alveograph: It was used the adapted hydration conditions determined for each sample with the Consistograph? Tests were carried out at 14% moisture basis?
  6. Bread-making: How much flower (wheat + germinated bean) was used to prepare each bread? Was the kneaded dough divided in several dough samples before fermentation? If so, which was the weight of each dough sample? Which was the cooking temperature? It was used tap or distilled water?
  7. Color: Please use CIE Lab* instead of CIE Lab.
  8. TPA: Which was the probe used? Which was the thickness of each bread slice? How many slices and repetitions in each slice were used? Which was the crosshead speed and time between each compression cycle?

Results

    1. Tables 1 and 2 and respective test: There are not significant differences (p >05) between means for all Consistographic and Alveographic parameters! Looking at some mean values and respective standard deviation, it seems very strange to me, I think results from statistics are not correct… Authors should check if these results are correct, this is of major importance… The same for all the other tables…
    2. If there are not significant differences (p > 0.05) between mean values, is not correct to comment about decrease/increase with BGF level… For example, in lines 198/203 “As it may be noticed, the water absorption value, dough consistency after 250 and 450 s decreased with the increased level of BGF addition in wheat flour. Regarding dough tolerance to mixing its value increased for dough samples up to 15% BGF addition followed by a slightly decreased. However all the dough samples with BGF addition presented higher value for tolerance to mixing compared to the control sample”, and lines 271/274 “Regarding the elasticity of the bread samples, it can be concluded that an addition of more than 15% (a, b letters) decreased the value of the elasticity, compared to the control sample (b letter), without any BGF addition in wheat flour”. In this sense, all manuscript should be revised. This is also of major importance.
    3. Figure 1: Frequency values in the axis should be in logarithmic scale.
    4. Table 5: Please use “L*” instead of “L”, and “Color parameters” instead of “Color measurements”.
    5. Lines 295/303: To support this text it would be nice to include a photograph of each bread.
    6. Table 6: Firmness, gumminess and chewiness values are too high, I think they are presented in “g” and not in “N”!!! Please check and correct it. This is another table where looking to the mean values (firmness, for example) it is very strange why there are not significant differences. Once again, you can´t say that firmness was influenced by BGF!!!

Author Response

17 June 2021

Dear Referee,  

We would like to thank the referee for the close reading and for the proper suggestions. We hope that we provide all the answers to the reviewer’s comments.

Thank you very much for the recommendations to publish our paper entitled “Dough rheological properties, microstructure and bread quality of wheat-germinated bean composite flour”.

The present version of the paper has been revised according to the reviewer’s suggestions.             

We uploaded the corrected version of the article for which we used the red color for the addition text.

GENERAL COMMENTS:

Referee comments: It is important to know the impact of adding healthy ingredients in the dough properties and technological aptitude of baked goods.

The article contains new information, the aim of the work is clearly established, discussion and conclusions are well documented. Besides technological and sensory impact of adding germinated bean flour to wheat flour, it would be very interesting to evaluate the impact in terms of nutrition and bioactivity.

However, my big concern is related to sample size and statistical analysis. All data were made in duplicate and one-way analysis of variance (ANOVA) with Tukey’s test was used to determine the significant differences at 5% level. To apply Tukey test, one of the assumptions is that the groups associated with each mean in the test are normally distributed and other is that there is homogeneity of variance. In theory, sample size is not important for Tukey test, but power of these results is dependent on sample size. In my opinion, a sample size of two is usually unpublishable, since it increases the risk that obtained results are not valid. In addition, some statistical results obtained for several parameters are strange (letters obtained for loaf volume and porosity, for example).

Response: We would like to thank to the referee for his/her close reading of our manuscript data. We agree with the referee point of view. The statistical part was not a correct one. We used XLSTAT for data processing and it was not updated (the program gave us errors). We used now SPSS for data processing and we obtained other statistical significance. We revised all the manuscript from the statistical point of view (according to SPSS processing). We hope now that are oK our data from the statistical point of view.

 Abstract

    1. Why the Authors use the abbreviation BGF for Germinated bean flour, instead of GBF?
    2. Please use L* instead of L.

 Response: We changed and we used know the GBF symbol.

  1. Keywords: “germinated bean flour” instead of “germinated bean”.

Response: We changed and we used know germinated bean flour.

 The entire document must be revised due to inaccuracies in some words and the absence of spaces between words and between words and bibliographic references.

Response: It is a problem of computer. A part of the manuscript was written on a computer and other one on another. We revised.

Lines 36-69: Authors say, “Through germination, the enzymatic activity of grains are increased which has the effect of facilitating the digestion of compounds such as starch and proteins and therefore makes germinated legumes to be used successfully in food where enzyme activity is required”. It is of great value to explain the technological importance of enzymes, especially amylases, on the dough fermentation process.

Response: We completed now in the manuscript the technological importance of enzymes, especially amylases, on the dough fermentation process.

Lines 49 and 61: Please avoid “etc.”.

Response: We deleted etc from the manuscript.

Lines 49/52: About beans protein, something should be said about its aminoacids profile.

Response: We completed in the manuscript with more informations about beans aminoacids profile.

Lines 52/54: Authors say, “Beans also presents a high content of bioactive substances such as tannins, flavonoids, phenolic compounds and other antioxidants”. Tannins and flavonoids are phenolic compounds… Please use a reference to support this sentence.

Response: We would like to thank to the referee for his/her remark. He/she has right. We revised and we added reference to support this sentence.

Line 67: “This process increases the bioavailability of nutrients, such as amino acids like lysine which is deficient in wheat flour [32], minerals present in the beans such as magnesium, iron, calcium, vitamins in especially B complex 65 in which beans are rich [33], decreases the amount of antinutrient compounds, such as phytic acid [34], trypsin inhibitors [32], and activates hydrolytic enzymes in the grain [35]”. Please add what is in bold.

Response: We added.

Materials and Methods

The type of flour should be indicated since it has high impact on the rheology and technological properties of wheat flour. Is it type 650 (line 144)?

Response: Yes, it is. We added now this fact in the manuscript.

Were doughs/breads prepared in duplicate? This is not specified… If not, the standard error only represents analytical measurement errors, and weakens your statistical power for identifying significant differences.

Response: We completed now in the manuscript that was made in triplicate.

Number of replicates should be presented in all experimental sections.

Response: We completed now in the manuscript the number of each replicates under the each table.

Consistograph: Tests were made using an adapted hydration level to achieve 2200 ± 100 mm H2O? Tests were carried out at 14% or 15% moisture basis?

Response: We made the mention now in the manuscript that test were out at moisture basis

Alveograph: It was used the adapted hydration conditions determined for each sample with the Consistograph? Tests were carried out at 14% moisture basis?

Response: We made the mention now in the manuscript according to the requirements.

Bread-making: How much flower (wheat + germinated bean) was used to prepare each bread? Was the kneaded dough divided in several dough samples before fermentation? If so, which was the weight of each dough sample? Which was the cooking temperature? It was used tap or distilled water?

Response: We made the mention now in the manuscript according to the requirements.

Color: Please use CIE Lab* instead of CIE Lab.

Response: We used now in the manuscript CIE Lab* instead of CIE Lab.

TPA: Which was the probe used? Which was the thickness of each bread slice? How many slices and repetitions in each slice were used? Which was the crosshead speed and time between each compression cycle?

Response: We completed in the manuscript how the TPA was made.

Tables 1 and 2 and respective test: There are not significant differences (p >05) between means for all Consistographic and Alveographic parameters! Looking at some mean values and respective standard deviation, it seems very strange to me, I think results from statistics are not correct… Authors should check if these results are correct, this is of major importance… The same for all the other tables…

Response: We revised the statistical part of the manuscript using SPSS.

If there are not significant differences (p > 0.05) between mean values, is not correct to comment about decrease/increase with BGF level… For example, in lines 198/203 “As it may be noticed, the water absorption value, dough consistency after 250 and 450 s decreased with the increased level of BGF addition in wheat flour. Regarding dough tolerance to mixing its value increased for dough samples up to 15% BGF addition followed by a slightly decreased. However all the dough samples with BGF addition presented higher value for tolerance to mixing compared to the control sample”, and lines 271/274 “Regarding the elasticity of the bread samples, it can be concluded that an addition of more than 15% (a, b letters) decreased the value of the elasticity, compared to the control sample (b letter), without any BGF addition in wheat flour”. In this sense, all manuscript should be revised. This is also of major importance.

Response: We revised the statistical part of the manuscript using SPSS.

Figure 1: Frequency values in the axis should be in logarithmic scale.

Response: We would like to ask the evaluator to allow us to mantain the original figure. We used the logarithmic scale but the figures are not so clearly ones in our opinion. See below.

Table 5: Please use “L*” instead of “L”, and “Color parameters” instead of “Color measurements”.

Response: We changed

Lines 295/303: To support this text it would be nice to include a photograph of each bread.

 Response: We added in the manuscript a photograph of each bread according to the referee suggestions.

Table 6: Firmness, gumminess and chewiness values are too high, I think they are presented in “g” and not in “N”!!! Please check and correct it. This is another table where looking to the mean values (firmness, for example) it is very strange why there are not significant differences. Once again, you can´t say that firmness was influenced by BGF!!!

Response: We revised in g for the firmness, gumminess and chewiness values. Also we revised the statistical part using SPSS.

Reviewer 2 Report

The article describes the effect of the fortification of bread with a flour obtained from germinated bean. Microscopic structure, technological properties, color and sensory properties of the composite bread were characterized. The topic is interesting, the experimental design was adequate and conclusions supported by the data.

Minor revisions are required before acceptance:

Please check through the text spaces between words often lacks;

Please justify the use of lyophilization to dry the germinated beans;

I suggest to express the loaf volume as specific volume;

I also suggest to provide statistical analysis of the results obtained in the sensory analysis.

A table including the nutritional data of the experimental composite breads will be useful for the readers.

Author Response

17 June 2021

Dear Referee,  

We would like to thank the referee for the close reading and for the proper suggestions. We hope that we provide all the answers to the reviewer’s comments.

Thank you very much for the recommendations to publish our paper entitled “Dough rheological properties, microstructure and bread quality of wheat-germinated bean composite flour”.

The present version of the paper has been revised according to the reviewer’s suggestions.             

We uploaded the corrected version of the article for which we used the red color for the addition text.

COMMENTS:

Referee comments:

The article describes the effect of the fortification of bread with a flour obtained from germinated bean. Microscopic structure, technological properties, color and sensory properties of the composite bread were characterized. The topic is interesting, the experimental design was adequate and conclusions supported by the data.

Response: We would like to thank to the referee for the his/her appreciation and the close reading of our manuscript.

Minor revisions are required before acceptance:

Response:

Please check through the text spaces between words often lacks;

Response: It is a problem of computer. A part of the manuscript was written on a computer and other one on another. We revised.

Please justify the use of lyophilization to dry the germinated beans;

Response: We justify now.

I suggest to express the loaf volume as specific volume;

Response: We changed in the manuscript loaf volume with specific volume.

I also suggest to provide statistical analysis of the results obtained in the sensory analysis.

Response: We changed now the way in which we provided the sensory analysis part. We provide the data with statistical analysis now.

A table including the nutritional data of the experimental composite breads will be useful for the readers.

Response: We included now in the manuscript a table with nutritional data for our bread samples.

Round 2

Reviewer 1 Report

The manuscript has been substantially improved by the Authors, following the recommendations. Now they used SPSS for data processing and statistical significance was obtained to support the discussion. The Authors also included data and the correspondent discussion about bread´s nutritional composition. The discussion section is very well supported by findings reported by other researchers.

My comments and suggestions for improvement concerns the following points:

  1. Figure 1: The graphics with logarithmic scale are not so clear because frequency sweep tests were performed from 1 to 20 Hz, a higher frequency range is usually applied to obtain the mechanical spectra. Yes, Authors could maintain the graphics without log scale.
  2. L302: “Regarding the elasticity of the bread samples, it can be concluded that an addition of more than 15% decreased the value of the elasticity, compared to the control sample, without any GBF addition in wheat flour”. Instead of “more than 15%”, please use “20% or 25%”.
  3. L312: “The addition of germinated bean flour to the bread-making recipe had the effect on all three physical data analyzed of the bread samples. A GBF addition up to 15% had the effect of improving their values”. Attention, this is not true in the case of elasticity values, 5-15% GBF breads are not different from control.
  4. Figure 4: As suggested, authors included a new figure with breads´ photos. However, the photographs of the loaves were taken from an angle that does not allow comparing the size of the slices. Moreover, apparently the volume of each bread does not correspond to the specific volume presented in the Table 4. For this reason, and if it is not possible to change the photos, I suggest deleting this figure.
  5. Table 6: Firmness, gumminess and chewiness values must be presented in Newton (S.I. unit).
  6. L346: “It was observed that the value of the firmness parameter was much higher than of the control sample after the addition of 15% GBF in wheat flour”. Please review this because there is no significant difference between control and 15% GBF.
  7. L348: “The gumminess and chewiness parameters had lower values than the control one up to an addition of 20% GBF in wheat flour”. The same here, with 20% GFB addition values are higher than control.
  8. Captions of Figures 3 and 5: F corresponds to 25% instead of 15% GBF.
  9. Table 7: “The results are mean ± standard deviation (n=3)”. However, “the sensory characteristics of the bread samples were evaluated by a panel of 20 semi-trained judges”. Therefore n=20, isn´t it?
  10. L490: “The increased values of temperature lead to an increased value of dough viscosity due to the starch gelatinization process. With increasing degree of starch gelatinization the dough viscosity decreased”. Please use “viscoelastic moduli” or “viscoelasticity” instead of “viscosity”. The same in L497, L502 and L678.
  11. L598: “The firmness of the bread was influenced by the surface viscosity of the lamellar liquid from the structure. Thus, with the decrease of the water absorption capacity of the mix flours by GBF addition, this parameter also changes. The more stable and firm structures are due to the higher surface viscosity, which leads to an increase in the firmness parameter”. In my opinion, Authors should not say it this way because surface viscosity of the lamellar liquid was not measured…
  12. L605: “Chewiness is a dimensionless size and characterizes the energy needed to chew the food and is characterized by the parameters: hardness, cohesiveness and springiness”. Please use “firmness” instead of “hardness”.
  13. L686: In the conclusions Authors say: “Crumb microstructure of breads samples shown larger pores size at high levels of GBF addition probably due to a more intense Maillard reaction”. However, in 4.3.4 section there is any reference to this… The impact of Maillard reaction should be related to color…

Author Response

27 June 2021

Dear Referee,  

We would like to thank the referee for the close reading and for the proper suggestions. We hope that we provide all the answers to the reviewer’s comments.

Thank you very much for the recommendations to publish our paper entitled “Dough rheological properties, microstructure and bread quality of wheat-germinated bean composite flour”.

The present version of the paper has been revised according to the reviewer’s suggestions.             

We uploaded the corrected version of the article for which we used the green color for the addition text.

GENERAL COMMENTS:

Referee comments: The manuscript has been substantially improved by the Authors, following the recommendations. Now they used SPSS for data processing and statistical significance was obtained to support the discussion. The Authors also included data and the correspondent discussion about bread´s nutritional composition. The discussion section is very well supported by findings reported by other researchers.

 Response: We would like to thank to the referee for his/her close reading of our manuscript data. We want to thank him/her for her/his support and comments. He/her help us a lot to improve our manuscript.

 My comments and suggestions for improvement concerns the following points:

  1. Figure 1: The graphics with logarithmic scale are not so clear because frequency sweep tests were performed from 1 to 20 Hz, a higher frequency range is usually applied to obtain the mechanical spectra. Yes, Authors could maintain the graphics without log scale.

Response: We would like to thank to the referee for let us mantain the original figures without log scale.

 L302: “Regarding the elasticity of the bread samples, it can be concluded that an addition of more than 15% decreased the value of the elasticity, compared to the control sample, without any GBF addition in wheat flour”. Instead of “more than 15%”, please use “20% or 25%”.

Response: We changed the formulation according to referee suggestions.

L312: “The addition of germinated bean flour to the bread-making recipe had the effect on all three physical data analyzed of the bread samples. A GBF addition up to 15% had the effect of improving their values”. Attention, this is not true in the case of elasticity values, 5-15% GBF breads are not different from control.

Response: We agree with the referee point of view. We revised the paragraph according to the referee suggestions.

Figure 4: As suggested, authors included a new figure with breads´ photos. However, the photographs of the loaves were taken from an angle that does not allow comparing the size of the slices. Moreover, apparently the volume of each bread does not correspond to the specific volume presented in the Table 4. For this reason, and if it is not possible to change the photos, I suggest deleting this figure.

Response: We do not have photo with breads from other angle. That way according to the referee suggestions we deleted the photos from the manuscript. We agree with the referee that the photo are not suggestive ones.

Table 6: Firmness, gumminess and chewiness values must be presented in Newton (S.I. unit).

Response: We presented the firmness, gumminess and chewiness values in Newton according to referee suggestions.

L346: “It was observed that the value of the firmness parameter was much higher than of the control sample after the addition of 15% GBF in wheat flour”. Please review this because there is no significant difference between control and 15% GBF.

Response: We revised the paragraph according to the referee suggestions.

L348: “The gumminess and chewiness parameters had lower values than the control one up to an addition of 20% GBF in wheat flour”. The same here, with 20% GFB addition values are higher than control.

Response: We revised the paragraph according to the referee suggestions.

Captions of Figures 3 and 5: F corresponds to 25% instead of 15% GBF.

Response: We would like to thank to the referee for his/her observations. We corrected.

Table 7: “The results are mean ± standard deviation (n=3)”. However, “the sensory characteristics of the bread samples were evaluated by a panel of 20 semi-trained judges”. Therefore n=20, isn´t it?

Response: We would like to thank to the referee for his/her observations. We corrected.

L490: “The increased values of temperature lead to an increased value of dough viscosity due to the starch gelatinization process. With increasing degree of starch gelatinization the dough viscosity decreased”. Please use “viscoelastic moduli” or “viscoelasticity” instead of “viscosity”. The same in L497, L502 and L678.

Response:  We changed according to the referee suggestions.

L598: “The firmness of the bread was influenced by the surface viscosity of the lamellar liquid from the structure. Thus, with the decrease of the water absorption capacity of the mix flours by GBF addition, this parameter also changes. The more stable and firm structures are due to the higher surface viscosity, which leads to an increase in the firmness parameter”. In my opinion, authors should not say it this way because surface viscosity of the lamellar liquid was not measured…

Response:  We revised the paragraph according to the referee suggestions.

L605: “Chewiness is a dimensionless size and characterizes the energy needed to chew the food and is characterized by the parameters: hardness, cohesiveness and springiness”. Please use “firmness” instead of “hardness”.

Response:  We corrected. We used “firmness” instead of “hardness” according to referee suggestions.

L686: In the conclusions Authors say: “Crumb microstructure of breads samples shown larger pores size at high levels of GBF addition probably due to a more intense Maillard reaction”. However, in 4.3.4 section there is any reference to this… The impact of Maillard reaction should be related to color…

Response:  We would like to thank to the referee for his/her point of view. We agree with his/her remark and we revised the paragraph.
